# Anxiety and Depression Among University Students in Cali, Colombia: A Cross-Sectional Analysis of the Prevalence and Contributing Factors

**DOI:** 10.3390/ijerph22091445

**Published:** 2025-09-18

**Authors:** Florencio Arias-Coronel, Carlos Andres Garibello-Millan, Diana María Dávila-Vidal, María Fernanda Serna-Orozco, Mauricio Solórzano-Alarcón

**Affiliations:** 1Faculty of Health, Universidad Santiago de Cali, Cali 760035, Colombia; carlos.garibello1@gmail.com (C.A.G.-M.); diana_davila4@hotmail.com (D.M.D.-V.); maria.serna05@usc.edu.co (M.F.S.-O.); mauricio.solorzano00@usc.edu.co (M.S.-A.); 2Grupo de Investigación Salud Integral (GISI), Universidad Santiago de Cali, Cali 760035, Colombia; 3Grupo de Investigación en Biomateriales y Biotecnología (BEO), Universidad Santiago de Cali, Cali 760035, Colombia

**Keywords:** depression, anxiety, university students, sociodemographic, sexual orientation and parental education

## Abstract

Depression and anxiety are mental health conditions that can significantly impact individuals’ well-being, with various risk factors contributing to their severity. This study aimed to characterise the prevalence of anxiety and depression among a university population and examine the associated sociodemographic factors. Methods: A descriptive cross-sectional study was conducted with 394 students from a higher education institution. Sociodemographic factors were analysed using two validated instruments: the Hamilton Anxiety Scale and the PHQ-9 (Depression Symptom Test). The statistical analyses included Pearson’s chi-square test and Fisher’s exact test to assess associations. Results: The mental health outcomes were significantly influenced by several factors. Key variables associated with anxiety and depression included academic overload (* *p* < 0.001), gender (* *p* < 0.001), father’s education level (* *p* < 0.001), socioeconomic status (* *p* < 0.001), and sexual orientation (* *p* < 0.001). These findings highlight the complex interplay between sociodemographic factors and mental health in university students. Conclusions: Early identification and interventions for anxiety and depression should be considered to provide timely and relevant mental health programmes for university students.

## 1. Introduction

Mental health refers to a person’s emotional, psychological and social well-being. It affects how we think, feel, and act, and it influences how we handle stress, relate to others, and make decisions. Mental health is important at all stages of life, from childhood and adolescence to adulthood [1].

Depression and anxiety are mental health conditions with which university students may struggle during their learning process in any area of knowledge [2,3]. Both conditions are negative and are public health problems. Depression is evidenced by the psychosomatic conditions of sadness, low energy and loss of interest in carrying out activities sustained over time and with a high degree of impact on functionality [4]. Anxiety is associated with the psychological dimension that affects the emotional and behavioural systems, which can be reflected in symptoms such as excessive worry, agitation and fear with exaggerated responses to everyday situations [5,6].

It is expected that approximately 14% to 29% of the world’s population may suffer from anxiety at some point in their lives [6,7], and depression is a frequent cause of disability worldwide, affecting more than 264,000,000 people [7,8]. In the university population, about 39% (2489 people) of students presented some anxiety symptoms, and in many cases, these were not identified. As a consequence, timely attention was not paid to the students [8,9]. Compared to the general population of the same age, university students have presented higher rates of anxiety and depression in recent years [10].

National studies identified symptoms of anxiety in 56% and depression in 32% [11], and factors associated with the responsibilities assumed upon entering higher education were identified, in addition to contextual characteristics such as economic resources, sexual orientation, management of free time, and interpersonal relationships [11,12,13]. These figures grew significantly after the COVID-19 pandemic.

Multiple studies have been conducted in recent years to characterise university students’ depression and anxiety in relation to their academic performance, particularly during the COVID-19 pandemic. Post-COVID-19 research showed a prevalence of approximately 50% on average for depression in health science students [14,15,16], generally associated with age factors (18 to 21 years), low social support, and early years of study [17]. Anxiety is related to factors such as sleep quality and female gender [18]. Understanding the relationship between economic factors, pre-existing mental health conditions (anxiety and depression), gender, and parental education levels among university populations remains an important research gap. Establishing these associations is crucial to enhance mental healthcare provision for this demographic and enable timely interventions to support academic performance. Furthermore, more comprehensive investigation of these relationships is needed.

The Beck Depression Inventory (BDI) and the Generalised Anxiety Disorder Scale (GAD-7) are widely used instruments with strong psychometric properties, including high internal consistency (Cronbach’s α > 0.85) and test–retest reliability, and have been validated in diverse populations, including Latin American university students. The BDI assesses the severity of depressive symptoms across cognitive, affective, and somatic dimensions, while the GAD-7 measures generalised anxiety symptoms and their functional impact. Both scales can also serve to identify the presence of risk factors for depression and anxiety in academic contexts.

In Latin America, recent studies reported prevalence rates of depression among university students ranging from 20% to 35%, and anxiety rates exceeding 40%. In Colombia, national surveys have identified higher prevalence in students from lower-income households, women, and those whose parents have a lower education level. These sociodemographic and economic factors, together with variables such as academic performance and parental education, are crucial for understanding the mental health outcomes in this population and for designing timely intervention strategies.

## 2. Methods

The design was a quantitative, descriptive and cross-sectional cohort study. Pearson’s chi^2^ test and Fisher’s exact test were used to assess the associations between all combinations of depression and anxiety with the sociodemographic variables.

### 2.1. Sample Selection and Characteristics

This study included a randomly selected sample of 394 students, drawn from a total population of 23,043 enrolled at the educational institution in 2023 (according to official records). The sample size was determined using a 95% confidence level, a 5% margin of error, and a 50/50 heterogeneity assumption to ensure representativeness.

### 2.2. Assessment Instruments

Anxiety levels were evaluated using the Hamilton Anxiety Scale, which demonstrated strong reliability, with Cronbach’s alpha coefficients of 0.79 and 0.92 [19]. Depressive symptoms were assessed using the PHQ-9 Depression Symptom Test, which also showed high reliability, with Cronbach’s alpha coefficients ranging from 0.86 to 0.89 [20].

Inclusion criteria: Active student enrolment status during the data collection period and voluntary participation confirmed through signed informed consent.

Exclusion criteria: Students who were not enrolled during the study period, those with incomplete responses to the questionnaires, or individuals who declined to participate after providing initial consent were excluded from the analysis.

Data collection was conducted via a Google Forms platform that incorporated (1) digital informed consent documentation, (2) descriptions of all the assessment instruments, and (3) sociodemographic questionnaires. All participation was voluntary.

Understanding of the scales: Prior to administering the Hamilton Anxiety Scale and PHQ-9, a pilot test was conducted with a small group of students to ensure the clarity and comprehension of the questions. Participants were also provided with a brief explanation of the scales’ purpose and structure, and contact information was offered for any queries during the process.

Statistical analysis was performed using R software (version 4.3.1; R Core Team, 2023). The following packages were used and are cited according to the recommended R citation format: psych [21] for the reliability indices, lavaan [22] for the validity indices, and tidyverse [23] for data manipulation.

### 2.3. Scale Reliability and Validity

The Hamilton Anxiety Scale showed high internal consistency (standardised alpha rho = 0.88; McDonald’s omega = 0.89). The PHQ-9 demonstrated similar reliability (standardised alpha rho = 0.87; McDonald’s omega = 0.88).

Confirmatory factor analysis supported the structural validity of both scales, with the fit indices falling within acceptable ranges (NNFI((Non-Normed Fit Index) = 0.92, GFI (Goodness of Fit Index)= 0.96, SRMR(Standardized Root Mean Square Residual) = 0.06 for Hamilton; NNFI = 0.91, GFI = 0.95, SRMR = 0.07 for PHQ-9).

To identify factors associated with anxiety and depression, two questionnaires were administered. Both instruments utilised a 4-point Likert-type scale, with the responses coded in ascending order (0 = lowest level, 3 = highest level).

The total score for each construct was computed by summing the values of all the items. The possible score range was 0–60 for anxiety and 0–30 for depression.

Severity cutoffs were established as follows. Anxiety: 0–14 = normal range, 15–29 = mild anxiety, 30–44 = moderate anxiety, 45–60 = severe anxiety. Depression: 0–5 = minimal depression, 6–9 = mild depression, 10–14 = moderate depression, 15–19 = moderately severe depression, ≥20 = severe depression.

Additionally, dichotomous variables were generated for the presence of anxiety and depression. A score above 14 on the anxiety scale and a score above 9 on the depression scale were considered to indicate the presence of these affective states (yes/no).

### 2.4. Ethical Considerations

This research was conducted in accordance with all the ethical principles enshrined in the UNESCO Declaration of 1999, Resolution 8430 of 1993, and it is considered that this research does not present a risk to the participants according to the previous resolution. All participants were made aware of the purpose of the study, the tests to be used and the results that those tests could yield, as well as the confidential use of the information and the security of the storage of it, in addition to the benefit and impact of the analysis of the results. All of this was made clear on the informed consent form. The right to withdraw from participation in the research and the right to withdraw one’s data were made explicit, even when the survey had already been completed, and all the protocols were accepted by the ethics committee under approval number 25.08.2023-8.

## 3. Results

### Sociodemographic Description

A total of 394 students from different faculties took part in this study. Students from the Faculty of Health represented 59.6% of the sample, followed by those from Business (10.9%), Law (7.4%), Basic Sciences (6.9%), Engineering (5.6%), Education (5.1%), and Humanities (4.1%).

Regarding the age distribution, 83.2% of the students were aged 18–24 years. The predominant sexual orientation was heterosexual (83.8%). In terms of marital status, 89.1% of participants were single, and 94.2% had no children.

In terms of socioeconomic status, 80.2% belonged to strata 2 to 4 and 82.7% lived in urban areas. With regard to the education level of the parents, 65.5% of mothers and 51.0% of fathers had a technical or university education. In addition, 52.8% of students perceived an excessive academic burden, and 68.3% reported never having been diagnosed with a mental health disorder such as anxiety, depression, or stress.

In relation to the presence of anxiety and depression in the sample studied, 63.2% of the participants experienced some level of anxiety, ranging from mild to severe. Specifically, 39.8% had mild anxiety, 17.5% had moderate anxiety, and 5.8% had severe anxiety, while the remaining 36.8% were in the normal range.

In terms of depression, 44.9% of the students showed levels of depression ranging from moderate to severe. A detailed analysis showed that 23.1% had moderate depression, 9.6% had moderate depression, and 12.2% had severe depression. In addition, 28.9% had mild depression and 26.1% had minimal depression.

Significant associations were observed between the sociodemographic variables and anxiety and depression (Table 1).

The presence of a previous mental health diagnosis (such as anxiety, depression or stress) was significantly associated with anxiety and depression, with *p*-values < 0.001 in both cases. Specifically, 43.8% of students with a previous diagnosis reported anxiety, while 73.6% reported depression. In contrast, only 11.0% of those without a previous diagnosis showed anxiety and 31.6% exhibited depression.

Among the students who perceived academic overload, 60.2% experienced anxiety and 53.4% experienced depression, compared to 40.0% and 35.5%, respectively, among those who did not perceive overload (*p* < 0.001 for both).

The anxiety prevalence was higher among females (70.1%) than males (49.6%) (*p* < 0.001). The depression prevalence was also higher among females (49.0%) than males (36.8%) (*p* = 0.028).

Regarding the fathers’ education level, the depression prevalence was higher among students whose fathers had lower education (*p* < 0.001), while no significant difference was observed for anxiety (*p* = 0.553).

The education level of the mother was not significantly associated with either anxiety (*p* = 0.286) or depression (*p* = 0.698). 

Students from lower socioeconomic strata (1 and 2) had a higher prevalence of anxiety (73.0% and 74.5%, respectively) and depression (54.1% and 59.8%, respectively) (*p* ≤ 0.004 for anxiety, *p* < 0.001 for depression).

Students identifying as bisexual, transgender, or ‘other’ reported the highest rates of anxiety (92.6%, 100%, and 100%, respectively) and depression (70.4%, 100%, and 100%, respectively) (*p* < 0.001 for both variables).

## 4. Discussion

The confidential surveys have allowed us to find out about the factors that are associated with depression and anxiety in university students from different fields of knowledge. The socioeconomic factor of low income exposes students to depression and anxiety, as has been shown in a recent study, which found a prevalence of over 70% for depression. However, it is not conclusive that anxiety is associated with low parental education [24], a correlation found in the present study.

Regarding academic stress, previous research has found similar results for anxiety and depression and high academic stress [25]. However, the prevalence of anxiety was higher than that of depression in university students in all previous studies [26,27]. It should be noted that most research has focused on health science faculties. The female gender is also a relevant factor associated with the prevalence of anxiety and depression in university students, with a strong correlation and a higher prevalence of depression [28].

In the present study, the rate of anxiety in women is high compared to depression. Other authors have found a triad: gender, academic stress and low social support [29].

Another factor that is present and of particular concern is a non-heterosexual sexual orientation (bisexual and transgender), for which the present study found a prevalence of 96–100% for anxiety and 70–100% for depression. Similar findings were reported by a study in 2023, but with a prevalence rate of 46% and 55% for depression and anxiety, respectively, in a heterosexual population. For this minority group, being in a romantic relationship during university seems to be a protective factor [30]. 

This study aimed to identify and validate factors associated with depression and anxiety in university students across various academic disciplines. The findings reveal significant associations, with observed prevalence rates of 44% for depression and 66% for anxiety—markedly higher than those reported in the prior literature [31,32]. Notably, these elevated levels of depression and anxiety are significantly linked to pre-existing mental health diagnoses before university enrolment. This underscores the need for universities to prioritise early mental health screening and intervention strategies for incoming students.

### 4.1. Socioeconomic Factors and Mental Health

A strong association was found between low socioeconomic status (SES) and increased risk of depression and anxiety. Students from lower-income backgrounds exhibited a higher prevalence of these conditions, aligning with recent studies reporting depression rates exceeding 70% in this demographic [33]. However, while the parental education level was initially considered a potential protective factor, the data did not support this hypothesis. Both students with parents lacking formal education and those with university-educated parents showed similarly high levels of depression and anxiety, suggesting that economic stability, rather than parental education, plays a more critical role in mental health outcomes.

### 4.2. Academic Stress and Gender Disparities

Consistent with previous research, this study confirmed a strong correlation between academic stress and heightened levels of anxiety and depression [34,35]. While most prior studies indicated that anxiety is more prevalent than depression among university students [36], some conflicting findings exist. For instance, Mihăilescu et al. (2016) reported higher depression rates linked to academic stress [37]. It is worth noting that much of the existing research has focused on health science students, potentially limiting the generalisability. Additionally, female students demonstrated higher susceptibility, with their depression and anxiety rates significantly exceeding those of their male counterparts. This aligns with prior studies identifying gender, academic stress, and low social support as interconnected risk factors [38,39]. The overrepresentation of female participants in this and similar studies may partly explain these findings, although further research is needed to explore the underlying causes.

### 4.3. Mental Health in Sexual and Gender Minority Students

A particularly concerning finding was the extremely high prevalence of anxiety (96–100%) and depression (70–100%) among bisexual and transgender students. In contrast, a 2023 study reported substantially lower rates (46% depression, 55% anxiety) in heterosexual student populations [40]. For sexual and gender minority students, being in a romantic relationship during university appeared to serve as a protective factor, highlighting the potential benefits of social support interventions tailored to this group.

### 4.4. Study Limitations

This research has several limitations. The sample was drawn from a single university, limiting the generalisability of the findings. The study did not investigate the underlying causes of academic stress or whether students had access to mental health support.

Moreover, the sample primarily consisted of young, heterosexual students without children, reducing the diversity and limiting extrapolation to broader student populations. Although significant associations were identified, the lack of multivariate statistical models prevented adequate control for confounding factors. These limitations highlight the need for future longitudinal studies with more diverse and representative samples, as well as causal analyses to develop more effective mental health interventions in academic settings. Despite these constraints, this study provides valuable data on the anxiety and depression prevalence among students, emphasising the need to address these issues in higher education.

## 5. Conclusions

Early identification and intervention strategies for anxiety and depression should be implemented to provide timely and relevant mental health programmes for university students. Particular attention should be paid to vulnerable groups, including women, students from lower socioeconomic backgrounds, and non-heterosexual students.

Universities should promote gender-sensitive education, strengthen social and emotional support tools, and continuously evaluate curricula to ensure that their study techniques meet the diverse learning needs of students. In addition, economic support policies should be articulated through governmental and institutional programmes to address the socioeconomic barriers that may affect academic success. Special consideration should be given to students whose parents have a low level of education.

The results show that depression and anxiety are significantly associated with sociodemographic factors such as gender, field of study, and family economic background. These findings highlight the need to implement prevention and support strategies within the university setting and suggest potential future research directions focusing on targeted interventions for students’ mental health. 

## Figures and Tables

**Table 1 ijerph-22-01445-t001:** Correlation of the sociodemographic variables with anxiety and depression.

Variable		Anxiety		Depression	
		Yes	No	Yes	No
Do you have any mental health diagnosis?	With diagnosis	(109) 43.8%	(140) 56.2%	(92) 73.6%	(33) 26.4%
	Without diagnosis	(16) 11.0%	(129) 89.0%	(85) 31.6%	(184) 68.4%
	*p*-value	Chi-square test, *p*-value ≤ 0.001		Chi-square test, *p*-value ≤ 0.001	
Do you perceive an excessive academic workload from teachers?	yes	(150) 60.2%	(99) 39.8%	(111) 53.4%	(97) 46.6%
	No	(58) 40.0%	(87) 60.0%	(66) 35.5%	(120) 64.5%
	*p*-value	Chi-square test, *p*-value ≤ 0.001		Chi-square test, *p*-value ≤ 0.001	
Sex	Female	(183) 70.1%	(78) 29.9%	(128) 49.0%	(133) 51.0%
	Male	(66) 49.6%	(67) 50.4%	(49) 36.8%	(84) 63.2%
	*p*-value	Chi-square test, *p*-value ≤ 0.001		Chi-square test, *p*-value = 0.021	
Father’s education level	None	(12) 66.7%	(6) 33.3%	(13) 72.2%	(5) 27.8%
	Primary	(25) 69.4%	(11) 30.6%	(22) 61.1%	(14) 38.9%
	Secondary	(91) 65.5%	(48) 34.5%	(59) 42.4%	(80) 57.6%
	Technician/technologist	(61) 64.2%	(34) 35.8%	(45) 47.4%	(50) 52.6%
	University	(60) 56.6%	(46) 43.4%	(38) 35.8%	(68) 64.2%
	*p*-value	Chi-square test, *p*-value = 0.553		Chi-square test, *p*-value = 0.010	
Mother’s education level	None	(3) 1.2%	(1) 0.7%	(3) 1.0%	(1) 1.0%
	Primary	(15) 6.0%	(10) 6.9%	(19) 6.5%	(6) 5.8%
	Secondary	(76) 30.5%	(31) 21.4%	(84) 28.9%	(23) 22.3%
	Technician/technologist	(78) 31.3%	(47) 32.4%	(88) 30.2%	(37) 35.9%
	University	(77) 30.9%	(56) 38.6%	(97) 33.3%	(36) 35.0%
	*p*-value	Fisher’s exact test, *p*-value = 0.289		Fisher’s exact test, *p*-value = 0.073	
Socioeconomic status	1	(27) 73.0%	(10) 27.0%	(20) 54.1%	(17) 45.9%
	2	(76) 74.5%	(26) 25.5%	(61) 59.8%	(41) 40.2%
	3	(73) 52.1%	(67) 47.9%	(48) 34.3%	(92) 65.7%
	4	(44) 59.5%	(30) 40.5%	(25) 33.8%	(49) 66.2%
	5	(25) 69.4%	(11) 30.6%	(20) 55.6%	(16) 44.4%
	6	(4) 80.0%	(1) 20.0%	(3) 60.0%	(2) 40.0%
	*p*-value	Fisher’s exact test, *p*-value = 0.004		Fisher’s exact test, *p*-value = <0.001	
Sexual orientation	Bisexual	(25) 92.6%	(2) 7.4%	(19) 70.4%	(8) 29.6%
	Heterosexual	(195) 59.1%	(135) 40.9%	(133) 40.3%	(197) 59.7%
	Homosexual	(18) 69.2%	(8) 30.8%	(14) 53.8%	(12) 46.2%
	Transsexual	(2) 100.0%	(0) 0.0%	(2) 100.0%	(0) 0.0%
	other	(9) 100.0%	(0) 0.0%	(9) 100.0%	(0) 0.0%
	*p*-value	Fisher’s exact test, *p*-value <0.001		Fisher’s exact test, *p*-value <0.001	

## Data Availability

The authors keep a digital version of the data collected at the following link: https://docs.google.com/presentation/d/1iSw0JYw0qpWXD2sgh9Zkd0-ltehbnsWd/edit?usp=drive_link (accessed on 30 December 2023).

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
