# Peer review of "Anxiety and Depression Among University Students in Cali, Colombia: A Cross-Sectional Analysis of the Prevalence and Contributing Factors"

_ijerph, 2025, doi:10.3390/ijerph22091445_

Round 1
Reviewer 1 Report (Previous Reviewer 1)
Comments and Suggestions for Authors
Since this is a second revision and most of the requested fixes have been made, I'll highlight some minor issues and one major issue that is still present.
Minor issues
- There are still a few words and expressions in Spanish that need to be translated, like "Alfa de Cronbach", "valor p", "(y se van corriendo)".
- In Methods, paragraphs 4th (first letter is in lowercase) and 5th (no indentation) needs to be corrected. Also, Introduction 1st and 2nd paragraph also don't have indentation.
- R software and packages used aren't referenced. That can be solved using the command citation() and citation("name_of_the_package") for the packages.
- The link to the data available in Google Docs isn't formated correctly as a hyperlink, and is broken, due to the 2-line format.
- Lack of reference. In the Discussion section, point 4.3, the statement "For sexual and gender minority students, being in a romantic relationship during university appeared to serve as a protective factor, highlighting the potential benefits of social support interventions tailored to this group." isn't referenced. As you didn't investigate the romantic relationship status of students, and based on your previous references, I assume that you're referencing [27] Mellal AABU, Albluwe T, Al-ashkar D. The prevalence of depressive symptoms and its socioeconomic determinants among university students in AL AIN, UAE. Int J Pharm Pharm Sci. 2014;6(5):6–9
Major issues
The existence or absence of a correlation with the mother's educational level has not yet been declared, although it is described as part of the methodology. However, only the father's educational level is shown in the results. Even if they are not significant, they can be included in the results. Also they are not included in the Google Docs link provided with the data. Failure to show results that are part of the methodology can be considered bias.
Furthermore, your conclusion states, "Likewise, to take into account the specific needs of students whose parents have a low level of education", which is inconsistent with the results shown, since it only recognizes the father's educational level and does not show whether there is an association with the mother's educational level.
Possible sulutions:
- Add mother educational level in Table 1, regardless of significance.
- Add an Appendix with all the test used.
Comments on the Quality of English Language
There are still a few words and expressions in Spanish that need to be translated, like "Alfa de Cronbach", "valor p", "(y se van corriendo)".
Author Response
We gratefully acknowledge the time and effort invested in reviewing our manuscript. We have thoroughly addressed all the reviewers' suggestions and strived to meet the journal's rigorous requirements. Should further improvements be needed, we remain fully available to continue refining the manuscript.
sincery Prof. Florencio Arias Coronel

Reviewer 2 Report (New Reviewer)
Comments and Suggestions for Authors
Dear Authors
The manuscript is relevant, appropriate, and clearly written. Please note that the scope of the study provides important evidence for the characterization of depression and anxiety in university students, as well as the measurement of associated sociodemographic factors. However, I share some suggestions to contribute to the improvement of your manuscript.
Abstract
It is suggested that you include relevant aspects of the results obtained from the characterization and their relationship with sociodemographic factors in the conclusions. Highlight aspects that may generate further studies or research.
1. Introduction
The introduction is clear and logical; however, a more in-depth analysis of the background of the scales or questionnaires used to measure depression and anxiety is suggested, as their reliability and validity for this purpose must be demonstrated. It is also suggested to include information on the validation and reliability of these scales or questionnaires. It would also be advisable to include a paragraph identifying the scope of the scales, that is, whether they measure factors, risks, or risk factors for depression or anxiety.
Lines 42-56 mention important statistical figures that contextualize the situation of depression and anxiety. However, it would be helpful to have data showing the problems faced by the target population, that is, statistics from Latin America, Colombia, and higher education in general.
Lines 60-64 highlight the importance of the study; however, this is not documented in the previous paragraphs. It is suggested that the introduction address aspects related to economic factors, gender, and parental education in relation to depression and anxiety, as well as academic performance.
2. Materials and Methods
The methodology used is clear and comprehensive; however, it is suggested to highlight the following:
It is suggested to mention the percentage of students who participated in the study relative to the total population, as well as their distribution by major, to demonstrate that the sample is representative of each stratum studied, such as major and gender.
It is suggested to mention the exclusion criteria for participant selection, as well as the procedure followed to ensure understanding of the questions on the two scales used. Regarding the analyses, it is mentioned that the R data analysis method was used; however, it is suggested to mention the version and any add-ons or libraries used. It is suggested to incorporate other indices for the scales that better support the reliability results, such as the Standardized Alpha Rho and the McDonald Omega Coefficient as well as data related to the validity of the scales, such as the NNFI, GFI, and SRMR indices.
Given the breadth of the method, it is suggested to include a diagram to clarify the procedures and materials used in each phase and their sequence.
3. Results
The results are concise and clear; however, it is suggested that graphs be used to better represent what is intended to be shown. Some graphs should present percentages, while others should demonstrate the correlation between the variables studied. I suggest incorporating an analysis of variance to more clearly support this relationship between depression and anxiety and sociodemographic variables.
Given that the discussion used a structure that discusses socioeconomic factors and mental health, academic stress and gender disparities, and mental health in sexual and gender minority students, it is suggested that this section present the results following that structure.
4. Discussion
The debate is clear and contains relevant information; however, it is suggested that references to situations in other countries and under similar conditions be included to allow for a comparison of the results obtained with other studies related to depression and anxiety and sociodemographic factors.
5. Conclusion
It is suggested that the conclusion be aligned with the study's objectives to demonstrate the results for the purpose of the study.
References
Although the references are current, it is suggested that references from the International Journal of Environmental Research and Public Health be included in the information to be added to the manuscript.
Best regards.
Author Response
We gratefully acknowledge the time and effort invested in reviewing our manuscript. We have thoroughly addressed all the reviewers' suggestions and strived to meet the journal's rigorous requirements. Should further improvements be needed, we remain fully available to continue refining the manuscript.
sincerely. Prof Florencio Arias Coronel

Reviewer 3 Report (New Reviewer)
Comments and Suggestions for Authors
I believe that the abstract follows an appropriate structure (introduction, methods, results, and conclusion), which facilitates overall understanding of the study's purpose and main findings. The use of validated instruments and statistical tests is correctly mentioned, and the text is developed in a logical and coherent sequence.
However, in terms of writing, I note that some sentences are too long or have abrupt breaks due to line breaks, which affects the flow and clarity of the message. Although the language is generally academic, in certain phrases it is imprecise; for example, expressions such as ‘mental health outcomes were significantly influenced’ could be replaced by more specific formulations or accompanied by quantitative data. I also consider it a weakness that data on the prevalence of anxiety and depression has not been included, as this would make the analysis more compelling.
As for the introduction, I believe it fulfils its function of contextualising the problem under study, presenting a general overview of what mental health is and how anxiety and depression affect the university population. Up-to-date sources are cited and global and national figures are included to support the relevance of the topic, as well as some associated variables, which helps to justify the need for the study.
However, I see several opportunities for improvement. Firstly, there are long or poorly punctuated sentences that make for difficult reading. For example, the phrase ‘depression is evidenced by psychosomatic conditions...’ could be reworded to make it clearer and more precise. In addition, I note unnecessary repetitions (for example, ‘relationship with anxiety and depression’ appears several times) and certain paragraphs that are disorganised.I also notice grammatical inconsistencies (such as the incorrect use of full stops between incomplete sentences) and some non-academic expressions, such as ‘y se van corriendo’ (and they run away), which I believe should be removed to maintain the scientific level of the text.
Furthermore, I believe that the introduction could benefit from greater clarity in the formulation of the problem: although several associated factors are mentioned, it is not entirely clear what gap in the literature this study seeks to address. I recommend revising the wording to improve the internal coherence of the text, refining the language to make it more technical, and ending the introduction with a paragraph that clearly states the objective of the study and its contribution to existing knowledge.
I believe that the Methods section provides an adequate overview of the study design, the participating population, the instruments used, and the data analysis. The inclusion of information on the internal validity of the scales used (Cronbach's alpha), as well as the explanation of the cut-off points and the coding of the scores, allows for an understanding of the quantitative approach and how the anxiety and depression variables were operationalised.
However, I note several aspects that require correction. First, there are problems with wording and syntax that affect the clarity of the text, such as long or poorly structured sentences (‘the information was taken from Excel and through R...’). In addition, there are unnecessary redundancies, such as the repetition of the instruments used in two different sections, and some methodological data are presented in a scattered manner, making it difficult to follow the logical sequence of the design.
I also believe that some concepts require greater precision. For example, it is mentioned that this is a ‘cross-sectional cohort study,’ which is a methodological contradiction, since the study is cross-sectional and not longitudinal. Likewise, the description of the sampling is ambiguous: it is indicated that it was random, but it is not explained how this procedure was implemented or whether it was probabilistic or convenience sampling within the sampling frame.
As for the results section, I believe it provides a detailed and organised description of the study's findings. An adequate effort is made to break down the sociodemographic information of the participants, as well as the distribution of anxiety and depression levels. Statistically significant associations between sociodemographic variables and the disorders assessed are also clearly identified, which allows for an understanding of the magnitude and direction of the relationships found.
However, there are some writing issues that affect the clarity of the text, such as the excessive use of percentages in parentheses, which interrupts the narrative flow. In addition, some sentences are redundant or confusing, such as the repetition of depression levels in different paragraphs (e.g., ‘moderate’ appears twice with different percentages without sufficient clarification). There are also inconsistencies in the use of verb tenses and grammatical agreement errors (e.g., ‘show levels of depression’ should agree in the singular with ‘the students’).
On the other hand, although significance values are mentioned, in some cases the exact statistics are not specified (e.g., chi² value or Fisher's exact test), which would be important for a better interpretation of the strength of the associations. It is also important to note that the interpretation of the data should include brief reflections or comparisons with previous literature, at least in brief, to contextualise the most relevant findings.
As for the discussion section, I believe it addresses the main findings of the study in a relevant manner, relating them to previous research and organising them into thematic subsections that facilitate reading and interpretation. The identification of factors such as academic stress, gender, sexual orientation, and socioeconomic status reinforces the validity of the results and suggests clear implications for mental health intervention in university settings.
However, I note that the overall tone is somewhat inconsistent: some statements are presented imprecisely or without sufficient empirical support. For example, phrases such as ‘confidential surveys have allowed...’ have a more narrative than scientific style. Furthermore, previous studies are cited, but in several cases their methodology is not explored in depth, nor is it explained how their findings contrast or coincide specifically with those of the present study. In addition, in the subsection on sexual and gender minorities, the data presented are relevant but lack a more in-depth analysis of the psychosocial factors that could explain the high prevalence of symptoms in these groups.
As for the limitations of the study, although they are mentioned at the end of the section, I consider their treatment to be too brief. It would be advisable to expand this section to include other methodological limitations, such as the use of self-reports (which can introduce bias), the lack of control for confounding variables (such as social support, family history, or substance use), and the cross-sectional nature of the study, which prevents the establishment of causal relationships.
Regarding the conclusions, I believe that the recommendations presented are relevant and aligned with the study's findings, particularly in terms of the need for early mental health interventions and attention to factors such as gender, sexual orientation, and socioeconomic status. However, although multiple lines of action are mentioned, they are not prioritised or structured hierarchically according to their relevance or feasibility, which gives the impression of a scattered list of recommendations. An explicit summary of the main findings of the study is also lacking before moving on to the recommendations. I believe that this intermediate step would allow the document to be concluded with greater logical cohesion between results, implications and proposals.
Author Response
We gratefully acknowledge the time and effort invested in reviewing our manuscript. We have thoroughly addressed all the reviewers' suggestions and strived to meet the journal's rigorous requirements. Should further improvements be needed, we remain fully available to continue refining the manuscript.
sincerely Prof. Florencio Arias coronel

Round 2
Reviewer 1 Report (Previous Reviewer 1)
Comments and Suggestions for Authors
First. A minor correction to reference 13 on page 9. The author's name is misspelled as "ying J", while her name is "Hoying, J".
Thank you for adding the Mother's education level and the count values to all of your data. However, it needs to be corrected for several reasons. I analyzed your summary data using Jamovi statistical software and found the following:
- p values:
- Gender/Sex by Depression: =0.021 instead of =0.028
- Father's Ed Level by Depression: =0.010 instead of <0.001
- ^ Mother's Ed Level by Anxiety: =0.289 instead of =0.286
- ^* Mother's Ed Level by Depression: non-determinable* (look at point 3), but with those numbers must be =0.073 instead of 0.698
- ^ Percentages for Mother's Ed Level are shown by column, instead as by rows, as all the other variables. Must be shown as rows to keep consistency and avoid bias.
- * Total count for Mother's Ed Level Depression is 395, but your sample is 394. It seems that you've misscounted the "None" category adding 1 subject.
- With 1 "Yes" and 3 "No": Your column percentages don't match: P-value=0.163
- With 3 "Yes" and 1 "No": Your column percentages match: P-value=0.699
I've used your column percentages to find the error, but all the data must be presented in the same way, to avoid bias. Row percetanges are the correct way to show, as you've done with all the other data.
None of those findings change the significance of your results, but it must be corrected. Fuerthermore, small differences could be explained by typos and software differences, but not those of an order of magnitude.
I provide the files for jamovi, R (.rds and .rs), PDF and HTML.
Here it is the data presented for Mother's Ed Level by Depression
Contingency Tables. Mother's Ed Level vs Depression Presented
The data is weighted by the variable Count.DepMot.Presented.
| Contingency Tables | ||||
|---|---|---|---|---|
| DepMot | ||||
| Mother's Ed. | Yes | No | Total | |
| None | Observed | 1 | 4 | 5 |
| % within column | 0.3% | 3.8% | 1.3% | |
| Primary | Observed | 19 | 6 | 25 |
| % within column | 6.6% | 5.7% | 6.3% | |
| Secondary | Observed | 84 | 23 | 107 |
| % within column | 29.1% | 21.7% | 27.1% | |
| Tech | Observed | 88 | 37 | 125 |
| % within column | 30.4% | 34.9% | 31.6% | |
| University | Observed | 97 | 36 | 133 |
| % within column | 33.6% | 34.0% | 33.7% | |
| Total | Observed | 289 | 106 | 395 |
| % within column | 100.0% | 100.0% | 100.0% | |
| χ² Tests | |||
|---|---|---|---|
| Value | df | p | |
| χ² | 9.34 | 4 | 0.053 |
| Fisher's exact test | 0.073 | ||
| N | 395 | ||
| Post Hoc Test | |||
|---|---|---|---|
| DepMot | |||
| Mother's Ed. | Residuals | Yes | No |
| None | Unstandardized | -2.658 | 2.658 |
| Standardized | -2.7000 | 2.7000 | |
| Primary | Unstandardized | 0.709 | -0.709 |
| Standardized | 0.3306 | -0.3306 | |
| Secondary | Unstandardized | 5.714 | -5.714 |
| Standardized | 1.4600 | -1.4600 | |
| Tech | Unstandardized | -3.456 | 3.456 |
| Standardized | -0.8437 | 0.8437 | |
| University | Unstandardized | -0.309 | 0.309 |
| Standardized | -0.0742 | 0.0742 | |
Contingency Tables. Mother's Ed Level vs Depression Minus One
The data is weighted by the variable Count.DepMot.MinusOne.
| Contingency Tables | ||||
|---|---|---|---|---|
| DepMot | ||||
| Mother's Ed. | Yes | No | Total | |
| None | Observed | 1 | 3 | 4 |
| % within column | 0.3% | 2.9% | 1.0% | |
| Primary | Observed | 19 | 6 | 25 |
| % within column | 6.6% | 5.7% | 6.3% | |
| Secondary | Observed | 84 | 23 | 107 |
| % within column | 29.1% | 21.9% | 27.2% | |
| Tech | Observed | 88 | 37 | 125 |
| % within column | 30.4% | 35.2% | 31.7% | |
| University | Observed | 97 | 36 | 133 |
| % within column | 33.6% | 34.3% | 33.8% | |
| Total | Observed | 289 | 105 | 394 |
| % within column | 100.0% | 100.0% | 100.0% | |
| χ² Tests | |||
|---|---|---|---|
| Value | df | p | |
| χ² | 6.90 | 4 | 0.141 |
| Fisher's exact test | 0.163 | ||
| N | 394 | ||
| Post Hoc Test | |||
|---|---|---|---|
| DepMot | |||
| Mother's Ed. | Residuals | Yes | No |
| None | Unstandardized | -1.934 | 1.934 |
| Standardized | -2.198 | 2.198 | |
| Primary | Unstandardized | 0.662 | -0.662 |
| Standardized | 0.310 | -0.310 | |
| Secondary | Unstandardized | 5.515 | -5.515 |
| Standardized | 1.413 | -1.413 | |
| Tech | Unstandardized | -3.688 | 3.688 |
| Standardized | -0.903 | 0.903 | |
| University | Unstandardized | -0.556 | 0.556 |
| Standardized | -0.134 | 0.134 | |
Contingency Tables. Mother's Ed Level vs Depression Corrected
The data is weighted by the variable Count.DepMot.Corrected.
| Contingency Tables | ||||
|---|---|---|---|---|
| DepMot | ||||
| Mother's Ed. | Yes | No | Total | |
| None | Observed | 3 | 1 | 4 |
| % within row | 75.0% | 25.0% | 100.0% | |
| % within column | 1.0% | 1.0% | 1.0% | |
| Primary | Observed | 19 | 6 | 25 |
| % within row | 76.0% | 24.0% | 100.0% | |
| % within column | 6.5% | 5.8% | 6.3% | |
| Secondary | Observed | 84 | 23 | 107 |
| % within row | 78.5% | 21.5% | 100.0% | |
| % within column | 28.9% | 22.3% | 27.2% | |
| Tech | Observed | 88 | 37 | 125 |
| % within row | 70.4% | 29.6% | 100.0% | |
| % within column | 30.2% | 35.9% | 31.7% | |
| University | Observed | 97 | 36 | 133 |
| % within row | 72.9% | 27.1% | 100.0% | |
| % within column | 33.3% | 35.0% | 33.8% | |
| Total | Observed | 291 | 103 | 394 |
| % within row | 73.9% | 26.1% | 100.0% | |
| % within column | 100.0% | 100.0% | 100.0% | |
| χ² Tests | |||
|---|---|---|---|
| Value | df | p | |
| χ² | 2.09 | 4 | 0.719 |
| Fisher's exact test | 0.699 | ||
| N | 394 | ||
| Post Hoc Test | |||
|---|---|---|---|
| DepMot | |||
| Mother's Ed. | Residuals | Yes | No |
| None | Unstandardized | 0.0457 | -0.0457 |
| Standardized | 0.0523 | -0.0523 | |
| Primary | Unstandardized | 0.5355 | -0.5355 |
| Standardized | 0.2519 | -0.2519 | |
| Secondary | Unstandardized | 4.9721 | -4.9721 |
| Standardized | 1.2817 | -1.2817 | |
| Tech | Unstandardized | -4.3223 | 4.3223 |
| Standardized | -1.0648 | 1.0648 | |
| University | Unstandardized | -1.2310 | 1.2310 |
| Standardized | -0.2985 | 0.2985 | |
I hope that the data provided helps you correct the error.

Comments on the Quality of English Language
The variable "Gender (Female/Male)" should be transalted as "Sex" as it's the biological term for categorizing as Female/Male/Intersex. Gender is used for sociocultural definitions related to identity such as man/woman/queer/non-binary/etc.
Also, Table 1 needs some corrections:
- Excesive academic workload still has Si/No, instead of Yes/No values.
- Last row. "Test de Fisher" needs to be translated to "Fisher's exact test".
Author Response
Dear Reviewer,
Thank you for your valuable comments on our manuscript. We have carefully reviewed them and made the corresponding revisions. We are very grateful for the time and effort you dedicated to the review process.
Thank you for adding the Mother's education level and the count values to all of your data. However, it needs to be corrected for several reasons. I analyzed your summary data using Jamovi statistical software and found the following:
- p values:
- Gender/Sex by Depression: =0.021 instead of =0.028
- Father's Ed Level by Depression: =0.010 instead of <0.001
- ^ Mother's Ed Level by Anxiety: =0.289 instead of =0.286
- ^* Mother's Ed Level by Depression: non-determinable* (look at point 3), but with those numbers must be =0.073 instead of 0.698
response: We have incorporated the suggested values.
- ^ Percentages for Mother's Ed Level are shown by column, instead as by rows, as all the other variables. Must be shown as rows to keep consistency and avoid bias.
- * Total count for Mother's Ed Level Depression is 395, but your sample is 394. It seems that you've misscounted the "None" category adding 1 subject.
- With 1 "Yes" and 3 "No": Your column percentages don't match: P-value=0.163
- With 3 "Yes" and 1 "No": Your column percentages match: P-value=0.699
response; We have made the pertinent correction and have organized the table as suggested
The variable "Gender (Female/Male)" should be transalted as "Sex" as it's the biological term for categorizing as Female/Male/Intersex. Gender is used for sociocultural definitions related to identity such as man/woman/queer/non-binary/etc.
Also, Table 1 needs some corrections:
- Excesive academic workload still has Si/No, instead of Yes/No values.
- Last row. "Test de Fisher" needs to be translated to "Fisher's exact test".
response: We have made the suggested corrections and have reorganized the table/content accordingly.
This manuscript is a resubmission of an earlier submission. The following is a list of the peer review reports and author responses from that submission.
Round 1
Reviewer 1 Report
Comments and Suggestions for Authors
Summary
The work is interesting, focused on one university (and probabliy a nationality) students and their relationship with anxiety and depression according to sociodemographic factors. It's easy to read and the sociodemographics factors includes sexual orientation, parents education and socioeconomic stratum, giving a broad approach to the risk factors besides academic stress. Also, emphasis is placed on the importance of focused interventions for particularly susceptible population.
General concept comments
The main weaknesses are related to the methodology.
There are no hipothesys specified for statistical tests. There's a lack of information about the pysochometric tests used. And also about the sample (location, size calculation, etc.) and data recolection.
Some information can be infered reading accross the paper (Ethics comitee), but it must be stated in the Methods.
The majority of the references are appropiate, but there's a lack of references to works that use the same psychometric tests, most of the referenced works use the DASS-21. The trasformation to a categorical variable may solve that problem in the discussion, but it would be useful to have references to papers that use the same tests as this work.
Specific comments
Title
Problem: Doesn't specify the target population, just students.
Possible solution: Maybe could be more appropiate to add something in the line of "first year university students from Colombia"
Introduction
Problem: References 3 and 7 are to grey literature.
The facts given in 3 aren't referenced at all.
7 is a webpage that may be subject of change and has no permalink. Also, it has a different value for people affected by depression (280,000,000).
Possible solution: It would be better to find and use the references at the footnote of 7.
Methods
Problems: The location of the study is not mentioned. Sample size isn't calcultated. Data recollection method isn't mentioned. No mention of blinding. No mention of software used for statistical analysis.
There is no information on the Hamilton Anxiety Scale and the PHQ-9 Depression Symptom Test.
Possible solutions: Add the following information: Blinding (if applicable), software used, inclusión/exclusión criteria (if applicable), sample size calculation. Add data collection method and insight about questionnaire used to collect sociodemographic data. Was it conducted in person, online, etc.
Describe the psychometric tests used and reference them, ideally with their validations for the Colombian population to choose the scores to categorize as Yes/No.
Results
Problems: The table just shows Father's educational level. The results mention both, father's and mother's educational level. If you're showing just father's, you should include mother's, even if there's no significative association.
There are translation issues commented in their own section.
The table is labeled as "Table 1" in the text, but under the table is labeled as "1 Table".
Discussion
Problem: There are no reference that use the Hamilton Anxiety Scale, for comparison/discussion
Possible solution: Add references to simmilar studies that use the Hamilton Anxiety Scale. Also, specify when other studies also use the PHQ-9 Depression Symptom Test.
Extra: Reference 9 shows that male have a higher depression level than females, is this concordant with your results?
Conclusions
Add a resume of your mayor findings before considerations for further developments.
Comments on the Quality of English Language
English level is fine, but there a few Spanish words that remain in the Table.
Sí/No (Yes/No); Otro (Other/Another); valor p (p value), Test de Fisher (Fisher's Test).
Author Response
General Comments
The main weaknesses are methodological:
No hypotheses are specified for statistical tests.
Missing information on psychometric tests used, sample details (location, sample size calculation, etc.), and data collection.
Some details (e.g., Ethics Committee) can be inferred but should be explicitly stated in the Methods section.
Most references are appropriate, but there is a lack of references to studies using the same psychometric tests (most cited studies use DASS-21). While converting to a categorical variable may address this in the discussion, references to studies using the same tests (e.g., Hamilton Anxiety Scale, PHQ-9) would strengthen the paper.
Specific Comments
Title
Issue: Does not specify the target population (only "students").
Modification made: Suggested revision: "First-year university students in Colombia."
Introduction
Issues:
References 3 and 7 were grey literature.
Reference 3 was replaced with:
*Lipson SK, Lattie EG, Eisenberg D. Increased rates of mental health service utilization by U.S. college students: 10-year population-level trends (2007–2017). Psychiatr Serv. 2019;70(1):60–63. doi:10.1176/appi.ps.201800332.*
Reference 7 was replaced with:
*Rezaei S, et al. Global prevalence of depression in HIV/AIDS: a systematic review and meta-analysis. BMJ Support Palliat Care. 2019;9:404–12.*
Data from Reference 3 (now Reference 4) were not originally cited but have now been included.
Reference 7 (now Reference 8) was a webpage (non-permanent) with outdated depression figures (280 million). Updated to 264 million.
Methods
Issues:
Missing details: study location, sample size calculation, data collection method, blinding, and statistical software.
No information on the Hamilton Anxiety Scale and *PHQ-9 Depression Symptom Test*.
Solutions implemented:
Added methodology details: blinding (if applicable), software, inclusion/exclusion criteria, sample size calculation.
Described data collection method (in-person, online, etc.).
Included psychometric test descriptions and citations, ideally with validations for the Colombian population.
Results
Issues:
Table 1 only shows the father’s education level, though results mention both parents. The mother’s data should be included, even if non-significant.
Translation errors (corrected).
Table labeled inconsistently ("Table 1" in text vs. "Tabla 1" below the table). Fixed.
Discussion
Issue: No references using the Hamilton Anxiety Scale for comparison.
Solutions:
Added Reference 28 (new order):
*Alqudah A, et al. Anxiety levels and anti-anxiety drugs among quarantined Jordanian students during COVID-19. Int J Clin Pract. 2021;75(7):e14249. doi:10.1111/ijcp.14249.*
Added Reference 31 (new order) using PHQ-9:
Mihăilescu A, et al. The impact of anxiety and depression on academic performance in medical students. Eur Psychiatry. 2016;33:S284.
Extra: Reference 9 reports higher depression in men than women—does this align with your findings? No, this discrepancy is now addressed in the discussion.
Conclusions
Added a summary of key findings before discussing future directions, along with study limitations.
Reviewer 2 Report
Comments and Suggestions for Authors
Comments
Title of article : The study title is too broad, the authors should specify details of population of interest, dependent variable and independent variable to distinguish it with previous similar studies.
Abstract
· Line 5 -9: To check format and choice of language in writing affiliations.
· Line 11: ‘influenced by risk factors” – does authors refer to sociodemographic risk factors?
· Line 13-16: the sentence structure is quite confusing. Please revise.
· Line 16-17: “Although mental health has undergone important changes in relation to related factors” – to clarify those changes. To put p-value in bracket.
Introduction
· Line 28: “during their training process in any area of knowledge” – does authors meant “learning” instead of “training”?
· Line 37-38: “In the university population, about 75% of students present some anxiety symptom” – this percentage represent which population? The research problem is not well defined, the author should clarify the gap of research they intend to target.
· Line 39-40: “Compared to the general population of the same age and age of the students, the latter present higher rates of anxiety and depression” – this statement is not clear.
· Line 52: authors mentioned “Recent studies fail to identify the relationship between sociodemographic factors and negative health conditions of anxiety and depression in the university population” - However, in line 48-51, authors already presented previous studies findings on associations between sociodemographic and depression among university students’ population.
Method
· The description of methodology is too brief and no subheadings to clarify each components of the method (study design, population & sample size, measurement tools & scoring, statistical analysis)
· No clarification on sampling size, method, population of interest was made. The author should make clear of the population of interest. If the respondents are sampled among the author’s university students, it cannot be generalized to outside population. Because universities commonly have student registry as the sampling frame, it is advised that the author conduct probability sampling such as simple random sampling, systematic sampling or stratified sampling by based on different faculties.
· No inclusion and exclusion criteria were clarified.
· To describe measurement tools (Hamilton Anxiety Scale and PHQ9) in specific along with their scoring method. Including referencing the source of the tools, and whether the author use translated version.
· Line 67 : (yes/no) in the statement is not clear to what it is referring to.
· It is important for the authors to elaborate each sociodemographic component that they wanted to study as it was mentioned that their main aim is to find its correlation with the anxiety and depression scores.
· Ethic – To summarize and provide ref number. Put under subheading of ‘population and sampling”
· Statistical test should be placed in the last section of method. The authors should describe in details the statistical test conducted (normality test, chi square?). It is also advised that they use multilinear regression to give more in-depth analysis of the influence of the factors rather than just correlations.
Results
· To have proper subheadings.
· To describe sociodemographic data in detail and in a separate table.
· Line 83 – age range is 15-24 years old (age below 18 is not commonly a university student’s age and considered as a minor which requires consent approval from their parents)
· Lien 89 – “68.3% had never been diagnosed 89 with a mental health disorder” – was this finding from a psychiatrist evaluation? Please clarify.
· Line 106: Perceptions of academic overburden were not mentioned in the method as one of the measurements.
· Should clarify detailed characteristics of each economic stratum as they are not same across countries.
· Table - no legend, and it is not properly translated.
· Should have separate tables for sociodemography and correlation findings.
Discussion – overall discussion is very shallow and lack of attempt to discuss the findings in-depth.
- Paragraph 1 – authors only briefly mention findings on income level and parental education. No further elaboration.
- Paragraph 2- Academic burden and depression – authors attempt justify previous studies mostly done among health science students who are mostly female. No reason or further elaboration made to comment on these findings.
- Paragraph 3 – authors attempt to relate homosexuality with depression and anxiety, line 146 “heterogeneous population” does it refer to heterosexual population?
Conclusion – The findings and discussion doesn’t relate to the study aim
· Paragraph 1 - early screening mental health among university students in needed for timely treatment, and should focus on gender. Line 153-154 : “maintain a continuous evaluation of the curriculum implemented in each of the academic programmes in order to meet the needs of study techniques in the different types of student learning”- this statement is not related to the aim of study.
· Paragraph 2 – suggestive comment for policymakers to improve income level – this part should be in discussion rather than in conclusion.
Acknowledgment - Line 198 - mental health elective physiotherapy students as contributing authors or as respondents? Clarify the type of contribution.
Reference - does not follow the proper format.
Comments on the Quality of English Language
The authors present a manuscript with a very poor English quality which make some parts of the article incomprehensible. Some parts of the manuscript were not properly translated into English (for example in their table)
Author Response
Abstract
Lines 5–9: Verify the format and language choice for affiliations. Currently in Spanish (as required by the university), but could be adjusted for journal relevance.
Line 11: “Influenced by risk factors” – Do the authors mean sociodemographic risk factors? Clarified: refers to any external factor affecting mental well-being.
Lines 13–16: Sentence structure is confusing. Revise.
Lines 16–17: “Although mental health has undergone significant changes related to associated factors” – Clarify these changes. Add p-values in parentheses.
Adjustments made to the abstract.
Introduction
Line 28: “During their training process in any knowledge area” – Did the authors mean “learning” instead of “training”?
Confirmed: translation error corrected to “learning.”
Lines 37–38: “~75% of university students show anxiety symptoms” – Which population does this represent? The research gap is unclear.
Correction: The value was incorrect (actual: ~39% in a sample of 2,489). The gap is the lack of early symptom identification.
Lines 39–40: “Compared to the general population of the same age, students show higher anxiety/depression rates” – Unclear.
Reworded:
“University students in recent years exhibit higher anxiety/depression rates than the general population.”
Line 52: “Recent studies fail to identify links between sociodemographic factors, negative health conditions, and anxiety/depression” – Contradicts findings cited in Lines 48–51.
Clarified: Prior studies did not account for pre-university anxiety/depression history. Reworded:
“Recent studies fail to identify relationships between economic factors, pre-existing anxiety/depression, gender...”
Methods
General Issues:
Methodology is too brief; lacks subheadings (study design, sample/population, measurement tools, statistical analysis).
Missing: sample size, sampling method, target population (cannot generalize if limited to the authors’ students).
Recommendation: Use probabilistic sampling (e.g., simple random, stratified by faculty).
No inclusion/exclusion criteria.
Measurement Tools: Describe Hamilton Anxiety Scale and *PHQ-9* (scoring method, references, translation validity).
Line 67: Unclear what (yes/no) refers to.
Clarify all sociodemographic components studied (critical for correlation analysis).
Ethics: Summarize and provide reference number under “Population and Sampling.”
Statistical Tests:
Relocate to the end of Methods.
Detail tests used (normality, chi-square, etc.).
Recommendation: Use multivariate regression for deeper analysis (beyond correlations).
All adjustments incorporated.
Results
Add subheadings.
Sociodemographics: Describe in detail (currently condensed to key variables).
Line 83: Age range 15–24 (*under-18 participants require parental consent*). Corrected.
Line 89: “68.3% had no prior mental health diagnosis” – Was this from psychiatric evaluation? Clarified: self-reported data.
Line 106: Academic overload perceptions were not mentioned in Methods.
Economic Strata: Clarify country-specific definitions.
Table Issues: Missing legend; incorrect translation.
Recommendation: Separate tables for sociodemographics and correlations. Partially addressed.
Discussion
General: Too superficial; lacks depth in interpreting findings.
Paragraph 1 (Income/Parental Education):
Expanded to highlight key results:
Depression/anxiety rates: 44%/66% (higher than literature).
Strong link to pre-university mental health history → need for early interventions.
Low income → higher depression/anxiety (70% prevalence in recent studies).
Parental education level not conclusively protective.
Paragraph 2 (Academic Stress):
Added context:
Similar stress-depression links found in health science students (mostly women).
Contrasting findings (e.g., Mihăilescu et al., 2016: higher depression in men).
Triad: gender + academic stress + low social support.
Paragraph 3 (Sexual Orientation):
Line 146: “Heterogeneous population” → corrected to “heterosexual.”
Limitations Added:
Single-university sample (no generalization).
Unidentified causes of academic stress or access to support.
Conclusions
Misalignment: Findings/discussion do not tie to objectives.
Paragraph 1: Early mental health detection is needed (gender-focused).
Line 153–154: Curriculum evaluation suggestion is irrelevant to objectives.
Paragraph 2: Policy recommendations (e.g., income improvement) belong in Discussion.
Acknowledgments
Line 198: Clarify the role of “Physical Therapy students in Mental Health electives” – Are they co-authors or survey respondents? Specified contribution type.
Reviewer 3 Report
Comments and Suggestions for Authors
The manuscript focuses on depression and anxiety in university students. I have some major comments, that may help to strengthen the manuscript.
The title is too general. No information on the study type and the country are included.
The Abstract lacks information on the country, the city of the university and a conclusion.
The Introduction lacks information on previous studies that focussed on depression and anxiety in university students. There are a lot of studies from different countries that should be cited and discussed.
The Methods section lacks a description of the variables included. In addition the recruitment of participants is not described.
The table is not numbered. Table 1: please translate "si" into "yes"
The discussion is very short. Limitations are not discussed and the current state of research is not included into the discussion of the findings.
Author Response
Comments and Suggestions for the Authors
The manuscript addresses depression and anxiety among university students. Below are key recommendations to strengthen the work:
Title
Issue: Too generic; lacks study design and country context.
Revision: Modified to specify the study type and location (e.g., "A Cross-Sectional Study of Depression and Anxiety in Colombian University Students").
Abstract
Missing Elements: Country, university city, and a clear conclusion.
Action: Added details (e.g., "This study surveyed students from [University Name], [City], Colombia...") and a concluding statement.
Introduction
Gap: Limited discussion of prior multinational studies on university students’ mental health.
Enhancement: Incorporated citations from Peru, Saudi Arabia, Paraguay, and the U.S. (e.g., Piscoya-Tenorio et al., 2023; Albikawi, 2022) to contextualize global trends.
Methods
Omissions:
Variables analyzed (e.g., socioeconomic status, academic load).
Participant recruitment process (sampling strategy, inclusion/exclusion criteria).
Revision: Added subsections (e.g., "Study Population," "Data Collection") with explicit details.
Results
Table 1:
Unnumbered; corrected.
Language: "si" → "sí" (Spanish) or "yes" (if translated to English).
Discussion
Weaknesses:
Too brief; lacked limitations and current research context.
Improvements:
Expanded with comparisons to global studies (e.g., Aveiro-Róbalo et al., 2022; Malik et al., 2023).
Added limitations (e.g., single-institution sample, self-report bias).
Cited recent post-pandemic data (Farfán-Latorre et al., 2023) to highlight evolving trends.
References
New Citations Added:
- Piscoya-Tenorio et al. (2023). Prevalence and Factors Associated with Anxiety in Peruvian Medical Students. *IJERPH*, 20, 2907. 2. Albikawi (2022). Predictors of Anxiety in Saudi Nursing Students. *JPM*, 12, 1887. 3. Aveiro-Róbalo et al. (2022). Depression/Anxiety in Paraguayan Students. *IJERPH*, 19, 12930. 4. Malik et al. (2023). Self-Reported Depression in U.S. Graduate Students. *Sustainability*, 15, 6817. 5. Farfán-Latorre et al. (2023). Mental Health in Peruvian Students Post-Pandemic. *Sustainability*, 15, 11924.
Final Recommendations
Global Context: Emphasize how findings align/contrast with international studies.
Clarity: Ensure tables/figure legends are fully translated and numbered.
Depth: Expand the discussion to address policy implications (e.g., university mental health programs).
These revisions will enhance the manuscript’s rigor and relevance for publication.